# Clinical validation of enhanced CT imaging for distal radius fractures through conditional Generative Adversarial Networks (cGAN)

Hyojune Kim[1], Seung Min Ryu[2], Ji-Soo Keum[3], Sang-Il Oh[3], Kyung-Nam Kim[3], Young Ho Shin[2], In-Ho Jeon[2], Kyoung Hwan Koh[2]*

1 Department of Orthopedic Surgery, Hospital of Chung-Ang University of Medicine, Dongjak-gu, Seoul, Republic of Korea, 2 Department of Orthopedic Surgery, University of Ulsan College of Medicine, Asan Medical Center, Seoul, Republic of Korea, 3 Waycen Inc., Seoul, Republic of Korea

* osdoc.koh@gmail.com

**Data Availability Statement:** Data cannot be shared publicly due to ethical considerations. Researchers who meet the criteria for access to confidential data can direct their data requests to

## Abstract

### Background/purpose

Distal radius fractures (DRFs) account for approximately 18% of fractures in patients 65 years and older. While plain radiographs are standard, the value of high-resolution computed tomography (CT) for detailed imaging crucial for diagnosis, prognosis, and intervention planning, and increasingly recognized. High-definition 3D reconstructions from CT scans are vital for applications like 3D printing in orthopedics and for the utility of mobile C-arm CT in orthopedic diagnostics. However, concerns over radiation exposure and suboptimal image resolution from some devices necessitate the exploration of advanced computational techniques for refining CT imaging without compromising safety. Therefore, this study aims to utilize conditional Generative Adversarial Networks (cGAN) to improve the resolution of 3 mm CT images (CT enhancement).

### Methods

Following institutional review board approval, 3 mm-1 mm paired CT data from 11 patients with DRFs were collected. cGAN was used to improve the resolution of 3 mm CT images to match that of 1 mm images (CT enhancement). Two distinct methods were employed for training and generating CT images. In Method 1, a 3 mm CT raw image was used as input with the aim of generating a 1 mm CT raw image. Method 2 was designed to emphasize the difference value between the 3 mm and 1 mm images; using a 3 mm CT raw image as input, it produced the difference in image values between the 3 mm and 1 mm CT scans. Both quantitative metrics, such as peak signal-to-noise ratio (PSNR), mean squared error (MSE), and structural similarity index (SSIM), and qualitative assessments by two orthopedic surgeons were used to evaluate image quality by assessing the grade (1~4, which low number means high quality of resolution).

### Results

Quantitative evaluations showed that our proposed techniques, particularly emphasizing the difference value in Method 2, consistently outperformed traditional approaches in

the Asan Medical Center Institutional Review Board. Contact details: Asan Medical Center, 88 Olympic-ro 43-gil, Songpa-gu, Seoul 05505, Korea. Tel: 02-3010-7166.

**Funding:** This work was supported by the National Research Foundation of Korea (NRF) grant funded by the Korea government(MSIT) (RS-2023-00278547), awarded to HK. The funder had no role in study design, data collection and analysis, decision to publish, or preparation of the manuscript.

**Competing interests:** NO authors have competing interests.

achieving higher image resolution. In qualitative evaluation by two clinicians, images from method 2 showed better quality of images (grade: method 1, 2.7; method 2, 2.2). And more choice was found in method 2 for similar image with 1 mm slice image (15 vs 7, p = 201).

## Conclusion

In our study utilizing cGAN for enhancing CT imaging resolution, the authors found that the method, which focuses on the difference value between 3 mm and 1 mm images (Method 2), consistently outperformed.

## Introduction

Accounting for approximately 18% of fractures in patients 65years and older, distal radius fractures (DRFs) are among the most commonly occurring fractures [1–3]. Although plain radiographs remain the gold standard for both diagnosis and classification, there's an increasing recognition of the value of computed tomography (CT) in obtaining detailed images crucial for precise diagnosis, prognosis, and intervention planning [4, 5]. Modern spiral CT scanners, despite their associated radiation dose increment, can furnish thin-slice, high-resolution images that achieve anatomical details with precision. High-resolution CT imaging is particularly essential for intricate cases where a nuanced understanding of fracture patterns, which directly influences therapeutic outcomes [6]. These preoperative CT scans play a pivotal role in preoperative planning, simulating fracture reduction, while intra-operative X-rays guide implant positioning during surgical procedures [7–10].

The advent of 3D printing in orthopedics marks a transformative era, especially evident in its applications for preoperative strategizing and crafting patient-specific implants [11–15]. For such applications, high-definition 3D reconstructions, predominantly derived from high-resolution CT scans, are imperative. In addition, the mobile C-arm CT, has emerged as an instrumental modality in orthopedic diagnostics and intraoperative imaging [16]. The C-arm CT, owing to its maneuverability and real-time imaging capabilities, empowers orthopedic surgeons with enhanced intraoperative visualization, facilitating precise interventions, especially in intricate procedures like osteotomies, joint replacements, and spinal surgeries [17–19]. However, despite its utility, the resolution of images obtained from certain C-arm CT devices can occasionally be suboptimal, potentially hampering accurate lesion characterization and surgical planning [19]. Therefore, there are previous researches to improve the quality and reconstruction of the image using multi-view images, extensively [20, 21].

Yet, the overarching concern of radiation exposure remains a deterrent, often curtailing the full exploitation of CT's potential. This caution stems from documented risks associated with prolonged ionizing radiation exposure, linking it to increased carcinogenic potentials [22–25]. In addition, because of the low quality of images examined with aging equipment, re-examination of the injured lesion is also increasing every year, and negative aspects such as increased exposure to the same area and increased medical costs are emerging as social problems. According to data released by the Health Insurance Review and Assessment Service in 2011, the rate of CT reexamination within one year was approximately 20% [26].

In the light of these challenges, our investigation embarks on a quest to refine CT imaging. Harnessing cutting-edge computational techniques, we aspire to enhance the resolution of C-arm CT images, mindful of mitigating radiation exposure. A successful outcome from this

endeavor promises to bolster the diagnostic and therapeutic tools available to orthopedic surgeons, offering sharper imaging insights without jeopardizing patient safety.

With the increasing interest in artificial intelligence and its applications in medical imaging, various methods have been proposed to enhance the resolution of images. Traditional image super-resolution techniques, such as bicubic interpolation, sparse coding, and self-example learning, have made significant contributions to the field [27–29]. More recently, deep learning-based approaches, like Convolutional Neural Networks (CNNs) and autoencoders, have been explored for their potential in achieving higher-quality image reconstructions with finer details [29]. Meanwhile of these advancements, this study specifically utilized conditional Generative Adversarial Networks (cGANs), a state-of-the-art deep learning framework that offers promising results in image-to-image translations [30–32]. Recognizing the potential of these advanced techniques, our research investigates a tailored approach to improve CT scan resolution, taking into account clinical implications and patient safety.

Therefore, the aim of this research is to evaluate the potential of conditional GANs in achieving superior CT image resolution without compromising patient safety. We hypothesize that by concentrating on the differential values between the original 3 mm input and the corresponding 1 mm images, the conditional GANs would yield enhanced results compared to traditional raw data-based.

## Materials and methods

### Data collection and preparation

This study was conducted after receiving approval from the institutional review board (AMC, No 2022–0210). The informed consent was obtained, and records from the CT database of a single center were collected, prospectively. The patients, who had CT examinations for distal radius fracture, were included from June 7[th], 2023 to December 31[st] 2023. We included patients' age from 19 to 80 years old. We evaluated the patients who had paired 3 mm-1 mm wrist CT scans that possible to be utilized for deep learning training and test. We excluded patients with (1) rheumatoid arthritis, (2) suspicion of infection, (3) metastatic lesion, (4) a history of operation at affected side, and (5) deformity of the wrist joint. After excluding these patients, a total of 11 patients were eligible and enrolled for this study. We scrutinized demographic data and assessed the fracture patterns based on the AO/OTA Classification [33].

From the collected data of 8 patients, 513 3 mm images and 2176 1 mm images, which paired images, were used for training, and 201 3 mm images and 1004 1 mm images of data from 3 patient were used for evaluation. In this study, we utilized paired 3 mm-1 mm CT data from patients who underwent a single CT examination at 1 mm slice thickness. The 3 mm images were virtually generated from these 1 mm scans to minimize radiation exposure, adhering to ethical standards and prioritizing patient safety. For training, CT data is used to optimize the deep learning model, and evaluation data is used to evaluate the degree to which 3 mm CT images are similar to 1 mm scan data generated by inputting them into the deep learning model. Additionally, a qualitative evaluation is conducted using 119 3 mm CT images of two patients collected at an external institution and images generated by a deep learning model.

### Network architecture and training

**Definitions.** "Thickness of CT slice" and "scanning interval" are core concepts for resolution surrounding CT imaging. Slice thickness refers to the axial resolution of a CT slice, whereas scanning interval refers to the distance between consecutive two slices. If slice thickness was smaller than scanning interval, there would be no information for the skipped section. Therefore, these two specifications have a same value in general. Fig 1 shows the simple

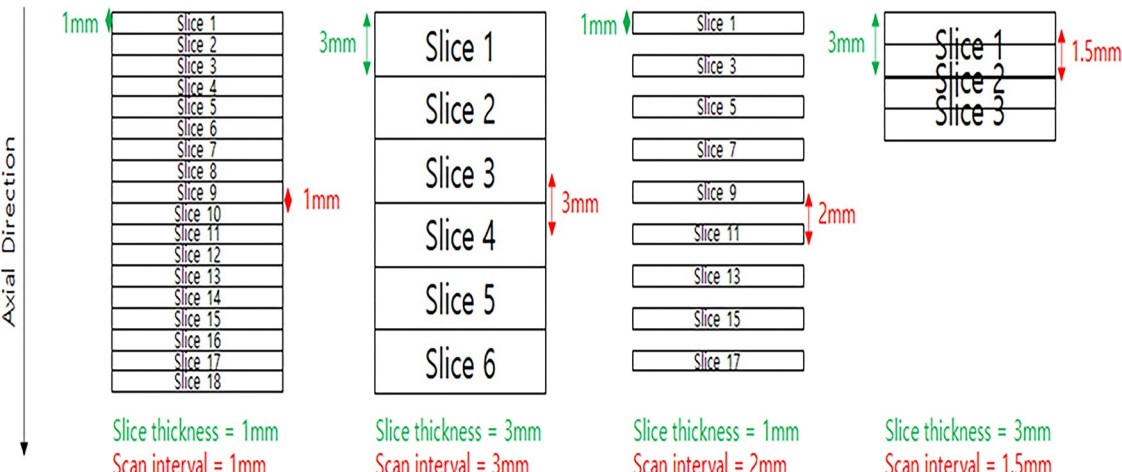

**Fig 1. The definitions of slice thickness and scanning interval.** Illustration of the relationship between CT 'slice thickness' and 'scan interval'. (Left): Set-up depicting 1 mm 'slice thickness' and an identical 1 mm 'scan interval'. (Center): Configuration illustrating 3 mm 'slice thickness' and an identical 3 mm 'scan interval'. (Right): Scheme showcasing a 3 mm 'slice thickness' with a varied 1.5 mm 'scan interval'. Green arrows highlight the 'slice thickness', while red arrows indicate 'scan intervals'. 'Slice thickness' defines the axial resolution, while 'scanning interval' refers to the gap between two consecutive slices.

illustration of relationship between slice thickness and scanning interval. To enhance the quality of CT scan data acquired from a low-resolution CT scanner, both slice thickness and scanning interval should be considered. In this study, 1 mm CT scans indicates the scans with 1 mm thickness and 1 mm interval.

**Flow of CT enhancement.** To improve the quality of CT scans, two common processes are used: reducing the size of a 3 mm image to a 1 mm image and interpolating from that 1 mm image. Since this process consists of two steps—reduction and interpolation—the quality of the final image could be compromised if errors occur in either step. Therefore, in this study, we developed a method to simultaneously reduce thickness and generate adjacent images from 3 mm to 1 mm images at once by introducing conditional GANs, one of the GAN methods. Since Generative Adversarial Networks (GANs) were proposed in deep learning technology, various models for generating data such as video and voice have been proposed [34–36]. Conditional GANs are a technology that enables image-to-image translation and can create a target image from an input image [37]. Fig 2 shows the configuration of conditional GANs. (A) in

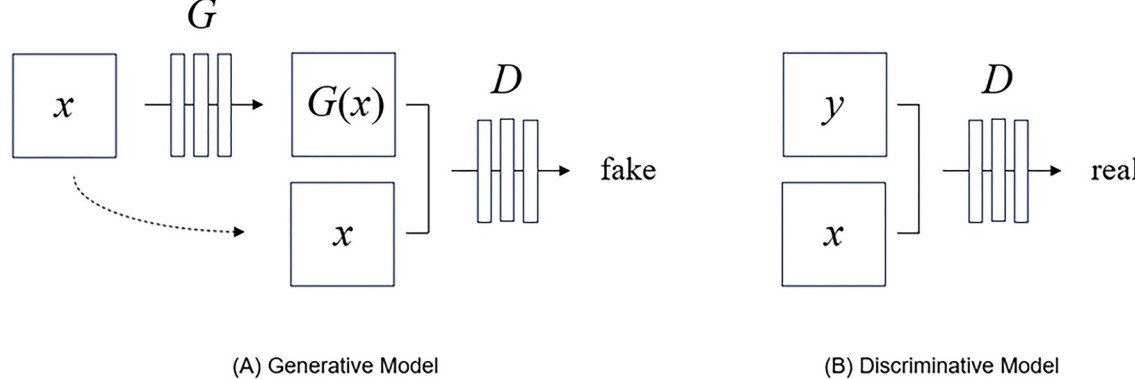

(A) Generative Model                    (B) Discriminative Model

**Fig 2. Conditional GANs model.** (A) It represents a generative model that generates a fake image from an input image (x). 'x' is the CT 3 mm input image. (B) It was a discriminative model that distinguishes the generated image into real and fake. 'y' is the 1 mm correct image. G, generative model; D, discriminative model; G(x), generated 1 mm CT image.

Fig 2 represents a generative model that generates a fake image from an input image, and (B) represents a discriminative model that distinguishes the generated image into real and fake. Our generator was 'U-Net'-based architecture [38], and for our discriminator we utilized a convolutional 'PatchGAN' classifier [39]. In the figure, x is the CT 3 mm input image, y is the 1 mm correct image, G is the generative model, D is the discriminative model, and G(x) represents the generated 1 mm CT image. The flow chart of the network was described in Fig 3.

Because the scan interval of a 1 mm CT image is shorter than that of a 3 mm CT scan image, differences may occur in the image corresponding to the 3 mm image. As a result of checking the difference image between the 3 mm CT image and the corresponding 1 mm CT image, changes could be confirmed depending on the rotation direction of the equipment. Fig 4 shows the difference image between the 3 mm CT input image and the 1 mm CT corresponding image. In the figure, (A) is a 3 mm input image, (B) is the corresponding 1 mm scan

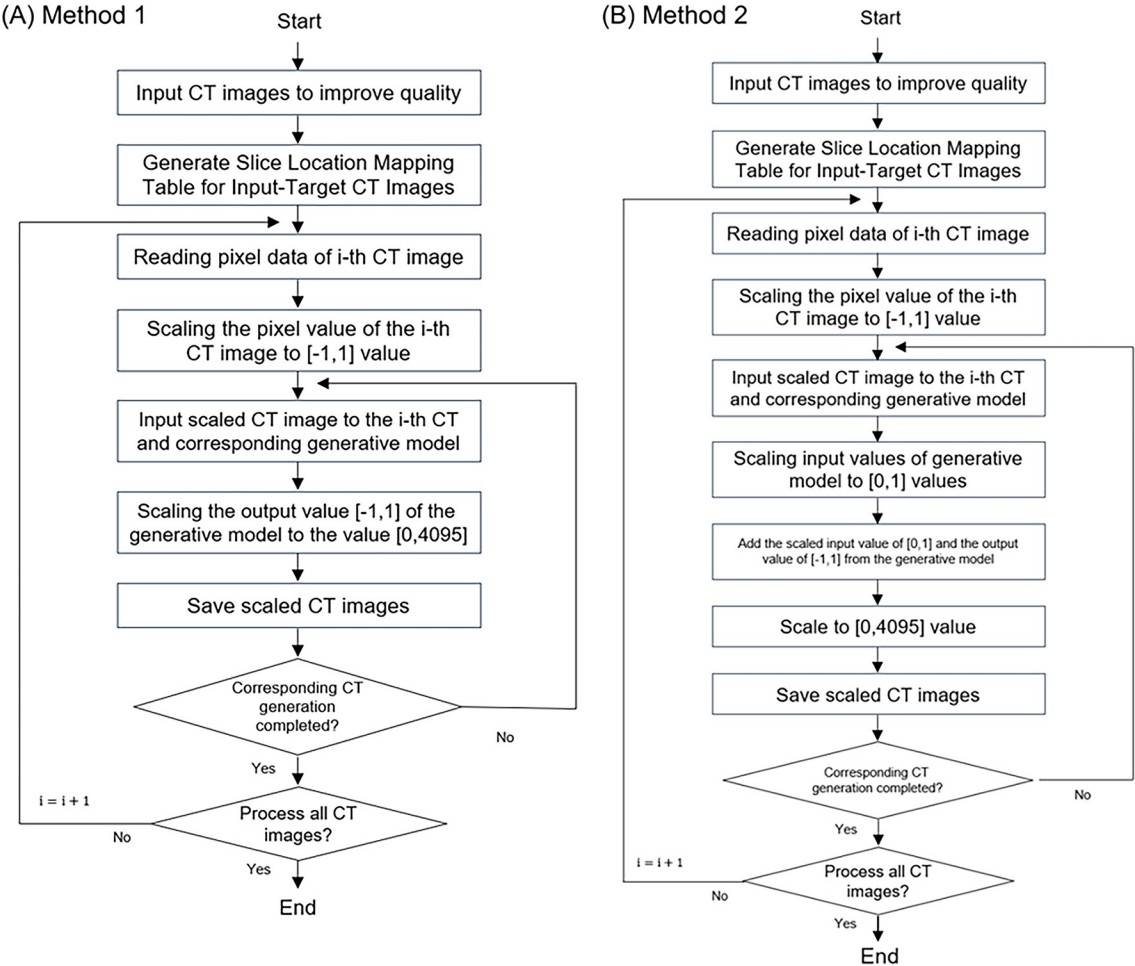

**Fig 3. Flow chart.** This flow chart illustrates the process of converting N sheets of 3 mm CT images into their corresponding 1 mm counterparts using two methods within a cGAN model. Each 3 mm CT image is processed individually and the operation is repeated N times. In Method 1 (A), the i-th 3 mm CT image's pixel value is read and input into the cGAN model in the range of [−1,1]. The model then generates an output that is scaled from the range [−1,1] to fit the CT image pixel value range of [0,4095]. Method 2 (B) also starts by reading the i-th 3 mm CT image's pixel value and inputting it into the cGAN model in the range of [−1,1]. However, this method involves an additional step where the scaled input value [0,1] is added to the output value [−1,1] of the generation model. The final step in both methods involves determining whether the 1 mm slices at positions 2P, 1P, C, 1N, and 2N, relative to the original 3 mm slice, have been created successfully. The generated 1 mm CT image is saved, and the process concludes when there are no more 3 mm CT images to be processed.

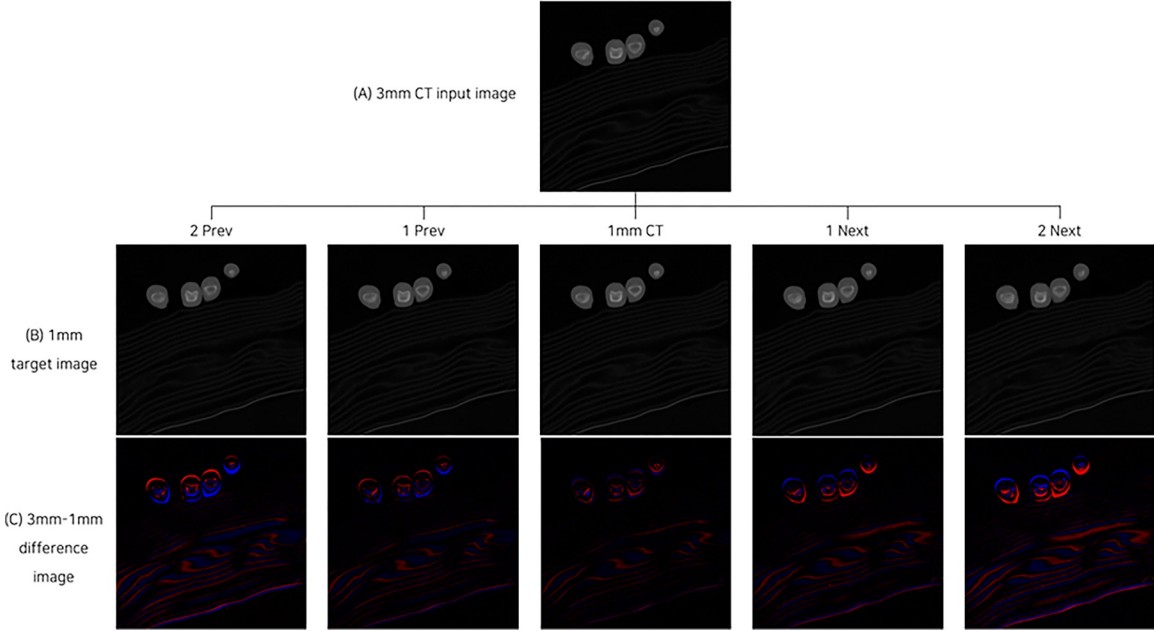

**Fig 4. Difference image between 3 mm CT image and 1 mm CT corresponding image.** (A) 3 mm input image. (B) the corresponding 1 mm scan image, and (C) is the difference image obtained by subtracting the 1mm image from 3 mm. The red pixel value means a positive value (3 mm is a bright pixel) and the blue pixel value is representing a negative value (1 mm is the brightest pixel). The difference between the 3mm image and the corresponding 1 mm CT image is the smallest in 1 prev. images and 1 mm CT. Otherwise, the difference between the adjacent before and after images is evident in 2 prev. and 2 next. images. Prev, previous.

image, and (C) is the difference image obtained by subtracting the 1 mm image from 3 mm. The red pixel value represents a positive value (3 mm is a bright pixel) and the blue pixel value is representing a negative value (1 mm is the brightest pixel). As can be seen in the second image, the difference between the 3 mm image and the corresponding 1 mm CT image is the smallest, and the difference between the adjacent before and after images is more evident.

As can be seen in the difference image in Fig 4, when generating a corresponding 1 mm CT image from a 3 mm CT image, it can be expected that quality improvement will be difficult if only one conditional GANs model is used for adjacent front and rear slices. Therefore, in this study, a method of constructing multiple models was applied to generate 1 mm CT images corresponding to 3 mm CT images. Fig 5 shows the configuration of the model for generating corresponding slices and adjacent slices.

**Training.** The optimization method proposed in the paper [37] was used to learn conditional GANs, which generate 1 mm CT images by inputting 3 mm CT images. In the 3 mm-1 mm paired images composed of training data, the 1 mm image corresponding to 3mm was used in the Typo, and the 1 mm images adjacent to 3 mm were used for training the 2P, 1P, 1N, and 2N models, respectively. It is noteworthy that there was a disparity in the intervals between the 3 mm images and the 1 mm images, with the former having an interval of 3 mm and the latter 0.7 mm. The grey column within the 3mm image denotes the matched area on the 1 mm image, as illustrated in Fig 5. Two methods were applied to training and generating CT images. 'Method 1' used a 3 mm CT raw image as the input image and was configured to generate a 1mm CT raw image. In 'Method 2', in order to focus on training the difference value between the 3 mm image and the corresponding 1mm image, the 3 mm CT raw image was used as the input image and the difference image value of the 3 mm and 1 mm CT was created (Fig 6). The "U-Net" was used, and 2D CT images were used for encoder, and produced as 2D CT [37].

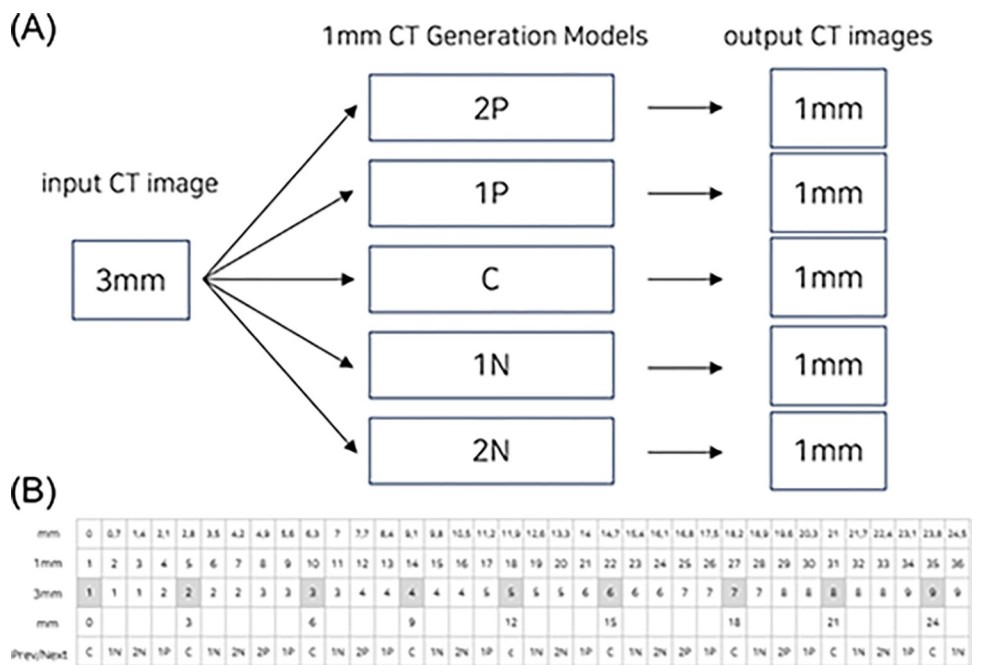

**Fig 5.** Configuration of 3 mm CT image input and multiple 1 mm generative models (A) and the example of slice location mapping table (B). The grey column within the 3 mm image means the matched area on the 1 mm image. The 1 mm image corresponding to 3 mm was used in the Typo, and 1 mm images adjacent to 3 mm were used for training the 2P, 1P, 1N, and 2N models, respectively. C, corresponding; 2P, before 2 slices; 1P, before 1 slice; 1N, after 1 slice; 2N, after 2 slices.

## Experiments

**Quantitative evaluation.**   We conducted a performance evaluation of the proposed method that simultaneously performs thickness reduction and interpolation of CT images. First, we confirmed the reproduction of the training data of the model that inputs the 3 mm CT image and generates the 1 mm CT image. By reproducing the training data, we can check whether the applied method is operating for its designed purpose. Next, cross-validation was conducted 3 times from the 8 patients, using 7 patient's data as training and 1 patient as testing. The test results of training data reproduction and cross-validation were compared with the image generated by inputting the 3 mm CT image and the correct 1 mm image.

**Metrics.**   There are several types of evaluation metrics that evaluate the similarity between two images, and each of them has its own unique characteristics. To evaluate CT quality improvement, the commonly used peak signal-to-noise ratio (PSNR), mean squared error (MSE), and structural similarity index (SSIM) were used [40]. Additionally, each evaluation index was applied to two methods, one that uses raw values of CT images and the other that focuses on training the difference values of corresponding images, to confirm the number of slices with improved quality for each patient.

**Qualitative evaluation.**   Two orthopaedic surgeons evaluated the set of images of CT using 2D axial serial images and 3D reconstructed images. The set was consists of 1 mm slice, 3 mm slice, and generated images from two different method 1 and 2. They made a grading from 1 to 4 for each image of set, in blinded. Grade 1 was highest resolution quality of images, and grade 4 was the lowest. The grading criteria were as follows:

## (A) Method 1

## (B) Method 2

**Fig 6.** Two models for generating 1 mm slice CT images from 3 mm images: (A) Method 1, (B) Method 2. (A) 'Method 1' used a 3 mm CT raw image (x) as the input image and was configured to generate a 1 mm CT raw image (G(x)). (B) 'Method 2' used 3 mm CT raw image (x) as the input image, and the difference values between 3 mm and 1 mm CT images (*) were trained. The different value was added to create 1 mm CT images (+). X, CT 3 mm input image; G, generative model; G(x), generated results.

- Grade 1: Excellent resolution with clear visualization of fine anatomical details for carpal bone and distal radius, including sharp cortical bone margins, fracture lines and distinct trabecular patterns.

- Grade 2: Good resolution with minor blurring but still adequate for clinical assessment of anatomical details for carpal bone and distal radius, but not exact for the fracture lines or trabecular patterns.

- Grade 3: Fair resolution with noticeable blurring, affecting the visibility of finer details, such as carpal bone and fracture patterns, but still allowing for basic clinical evaluation.

- Grade 4: Poor resolution with significant blurring, making it difficult to distinguish anatomical structures and unsuitable for clinical use for evaluating the fractures or anatomical structures.

Additionally, they were questioned to choose better quality images, which were similar to 1 mm slice images, between generated images from method 1 and 2. Two evaluators' agreement for grading and choice for better images between two generate images was assessed using Cohen kappa coefficients.

**Table 1. The results for reproducibility evaluation of training data.**

| | | Method 1 | | | Method 2 | | |
|---|---|---|---|---|---|---|---|
| | | PSNR | MSE | SSIM | PSNR | MSE | SSIM |
| Training | 1 | 49.29073 | 0.0000123 | 0.98783 | 49.86939 | 0.0000108 | 0.98762 |
| | 2 | 48.89408 | 0.0000135 | 0.98742 | 49.42825 | 0.0000119 | 0.98725 |
| | 3 | 52.76710 | 0.0000058 | 0.99509 | 53.93693 | 0.0000045 | 0.99560 |
| | 4 | 55.38347 | 0.0000034 | 0.99776 | 56.19667 | 0.0000029 | 0.99785 |
| | 5 | 52.12194 | 0.0000065 | 0.99539 | 53.16895 | 0.0000052 | 0.99541 |
| | 6 | 54.62009 | 0.0000041 | 0.99634 | 55.50990 | 0.0000034 | 0.99641 |
| | 7 | 52.00740 | 0.0000067 | 0.99509 | 52.97627 | 0.0000055 | 0.99540 |
| | 8 | 51.29487 | 0.0000079 | 0.99560 | 52.10701 | 0.0000065 | 0.99587 |
| mean | | 54.04746 | 0.0000075 | 0.99382 | 52.89917 | 0.0000063 | 0.99393 |

PSNR, peak signal-to-noise ratio; MSE, mean squared error; SSIM, structural similarity index

Grayish colored columns mean the better quality of images based on each metric (PSNR, MSE, SSIM) from the method 1 and 2.

# Results

Included patients' mean age was 64.8 years old. Six patients were injured at right side, and the other were left side. According to AO classification, all fractures were involved at intra-articular side (C2: 7 cases, C3: 4 cases).

## Quantitative results

The reproducibility evaluation results of the training data and the performance evaluation results of the test data for each evaluation scale are shown in Tables 1 and 2. Tables 3 and 4 show a performance comparison of a method that learns to generate raw CT images and a method that focuses on learning the difference value between the 3 mm input image and the 1 mm corresponding image. Table 3 showed that method 2 had more slices of images, which generated higher quality resolution, in PSNR, MSE metrics (85.1% vs 14.9%), and SSIM (63.6% vs 36.4%). In test data, the superiority was not very evident compared to training data (PSNR, MSE: 47.3% for method 2, SSIM: 65.8% for method 2).

Both proposed methods show high performance in training data reproduction evaluation, showing that thickness reduction and interpolation can be applied from a 3 mm CT image to a 1 mm CT image at once. In comparing methods for training data and test data, the method that focused on training the difference value between the 3 mm input image and the 1 mm corresponding image (method 2) showed higher performance compared to the method learning raw data (method 1).

## Qualitative results

The evaluated image's sample was in Fig 7. In two clinician evaluations for grading in each set, all images of 1 mm slice had the highest quality of the images (grade 1), and 3 mm slice images

**Table 2. The results for quantitative evaluation of test data.**

| | | Method 1 | | | Method 2 | | |
|---|---|---|---|---|---|---|---|
| | | PSNR | MSE | SSIM | PSNR | MSE | SSIM |
| Test | 9 | 47.22105 | 0.0000240 | 0.99313 | 46.45742 | 0.0000298 | 0.99212 |
| | 10 | 44.82873 | 0.0000454 | 0.98829 | 45.05453 | 0.0000433 | 0.98531 |
| | 11 | 44.74124 | 0.0000459 | 0.97661 | 44.89021 | 0.0000436 | 0.97714 |

Grayish colored columns mean the better quality of images based on each metric (PSNR, MSE, SSIM) from the method 1 and 2.

**Table 3. The results for performance comparison of two methods on training data.**

|  |  | Number of slices | PSNR, MSE | | SSIM | |
|---|---|---|---|---|---|---|
|  |  |  | Method 1 | Method 2 | Method 1 | Method 2 |
| Training | 1 | 215 | 34 | 181 | 120 | 95 |
|  | 2 | 233 | 58 | 175 | 97 | 136 |
|  | 3 | 353 | 15 | 338 | 51 | 302 |
|  | 4 | 391 | 66 | 325 | 156 | 235 |
|  | 5 | 208 | 20 | 188 | 79 | 129 |
|  | 6 | 302 | 65 | 237 | 148 | 154 |
|  | 7 | 225 | 41 | 184 | 77 | 148 |
|  | 8 | 249 | 26 | 223 | 65 | 184 |
| Total | | 2176 | 325 (14.9%) | 1851 (85.1%) | 793 (36.4%) | 1383 (63.6%) |

The number of each column for PSNR, MSE, SSIM means the number of slice, which has better quality of the image slice compared to the other method in each PSNR, MSE, and SSIM metrics.

were grade 4. Compared to images from generated by method 1, images from method 2 showed better quality of images (grade: method 1, 2.7; method 2, 2.2) (Fig 8). The kappa coefficient for method 1 was 0.792 (p = 0.007), and 2 for 0.744 (p = 0.011). And more choice was found in method 2 for similar image with 1 mm slice image (15 vs 7, kappa = 0.377, p = 201).

## Discussion

This study presents a comprehensive approach to enhance CT image quality using conditional GANs. The idea of simultaneously reducing the thickness and interpolating from a 3 mm CT image to a 1 mm image is novel and highly applicable in the clinical scenario, especially for intricate orthopedic cases such as distal radius fractures. Furthermore, we proved that the method 2, which focusing on the differential values between the original 3 mm input and the corresponding 1 mm images, produced more enhanced results in both ways of quantitative and qualitative ways than the method 1, which was traditional raw data-based approaches.

First, our results indicated that the feasibility of generating 1 mm CT images from 3 mm images by conditional GANs. In other medical departments, especially using chest CT, there were several trials to achieve high-resolution CT from low-resolution CT to increase detection the disease by deep learning method. They demonstrated their protocol as an acceptable by measurements of metrics of PSNR, SSIM (PSNR 32.60, SSIM 0.881) [41]. It could lessen the CT scanning time and decrease the radiation dose [41–43]. There were several suggested network models for enhancing the CT image quality: variants of GANs, and convolutional neural

**Table 4. The results for performance comparison of two methods on test data.**

|  |  | Number of slices | PSNR, MSE | | SSIM | |
|---|---|---|---|---|---|---|
|  |  |  | Method 1 | Method 2 | Method 1 | Method 2 |
| Test | 9 | 391 | 318 | 73 | 78 | 313 |
|  | 10 | 302 | 93 | 209 | 120 | 182 |
|  | 11 | 311 | 118 | 193 | 145 | 166 |
| Total | | 1004 | 529 (52.7%) | 475 (47.3%) | 343 (34.2%) | 661 (65.8%) |

The number of each column for PSNR, MSE, SSIM means the number of slice, which has better quality of the image slice compared to the other method in each PSNR, MSE, and SSIM metrics.

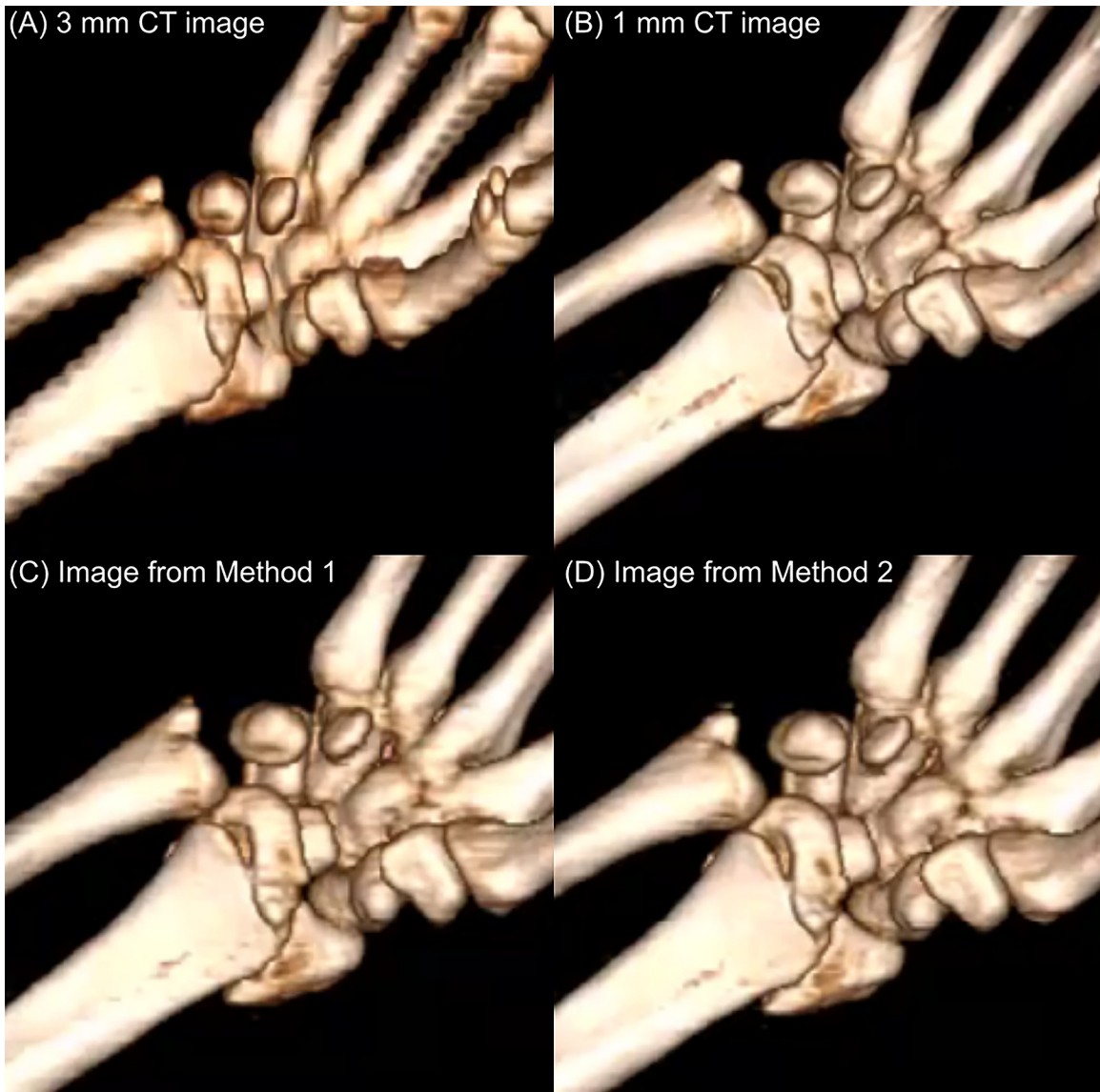

**Fig 7.** The evaluated sample of the set of images: (A) 3 mm CT image, (B) 1 mm CT image, (C) Image from Method 1, and (D) Image from Method 2.

networks (CNN). Among those networks, CNN is known to fail in the recognition of global structure in natural images, so it is less likely to preserve the needed anatomical structures for synthetic CT. However, GANs, because of its architectural and theoretical properties, can maintain the structural content of the images [43]. Conditional GANs have emerged as a versatile and effective solution for image-to-image translation challenges, demonstrating significant potential in applications such as enhancing the resolution of CT images. These networks are distinct in their ability to learn the mapping from input to output images while concurrently learning a loss function to train this mapping. This dual learning capability makes cGANs adaptable to a wide array of tasks that traditionally required unique loss formulations. A notable feature of cGANs is their use of the PatchGAN discriminator, which specifically targets high-frequency details in images. This approach is crucial in preserving the crispness and

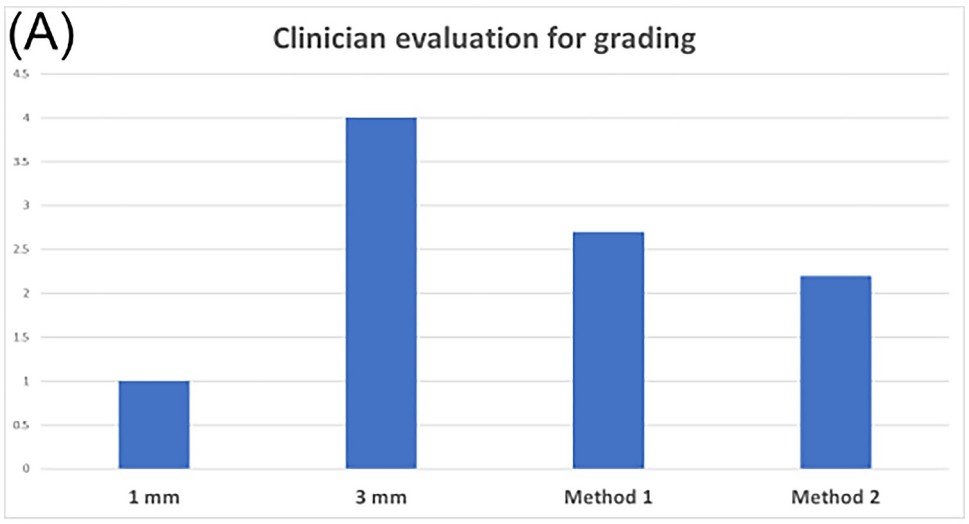

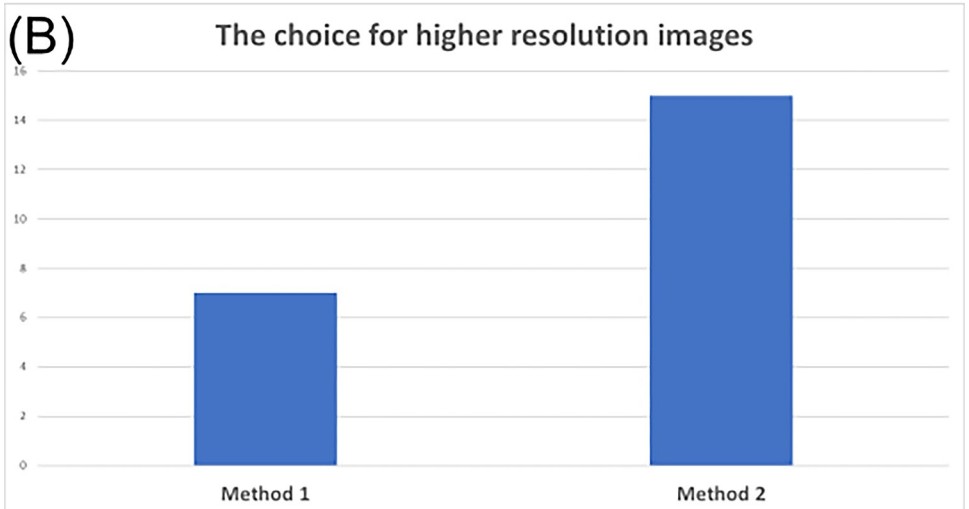

**Fig 8.** Quality evaluation for generated images from method 1 and 2: (A) grading for each image, and (B) the choice for better quality of image from method 1 and 2.

fine details in images, addressing the common issue of blurriness associated with L1 and L2 losses in image generation. In medical imaging, and particularly in CT image enhancement, the ability of cGANs to produce realistic images that maintain the structural integrity of the input is invaluable. The combination of L1 loss with the cGANs framework results in outputs that are not only realistic but also closely aligned with the input, ensuring that critical details and structures in medical images are accurately reproduced. This balance of realism and fidelity makes cGANs an ideal choice for enhancing the resolution of CT images, offering a promising direction for advancements in medical imaging technology. Therefore, we applied the GANs for deep learning.

While previous chest CT enhancement efforts serve as a precursor to our study, the challenges faced there are notably different. The respiratory, cardiac, and other motions introduce a variable that's absent in our orthopedic context. For instance, respiratory-induced motion artifacts in chest CT can distort the anatomical structures, making the interpolation task even more complex. Compared to their limitations, Our results, achieving a PSNR of 54.05 and

SSIM of 0.99, surpassed the metrics from chest CT research. This could be attributed to our controlled acquisition process, where both 3 mm and 1 mm images were obtained from a fixed patient's position, ensuring minimal external variances.

Secondly, we showed that the method focusing on the difference value between the 3 mm and 1 mm images (method 2) outperformed the raw data approach (method 1). The qualitative assessment further supported this notion, with orthopedic surgeons showing a preference for images generated by method 2. This method effectively provided a near-real representation of the 1 mm CT images, bridging the resolution gap. This implies that understanding the differential values can provide superior guidance for the enhancement algorithms. However, it was more evident in training data set, but not in test data. Deep learning models, when trained excessively on a specific dataset, tend to perform exceedingly well on that but might falter on unseen data. Regularization techniques, augmenting the dataset, or employing transfer learning can potentially ameliorate this discrepancy [44, 45].

Third, even compared to 3 mm slice, the reconstructed images from deep learning using method 1, 2 showed higher resolution, there was still limitations to improve the quality of 3 mm slice to 1 mm slice. In clinical settings, the keen eyes of radiologists and clinicians were able to distinguish the reconstructed images from the original 1 mm slices. This highlights the subtle intricacies and anatomical details present in genuine 1 mm slices that our current methods might not fully capture. Three primary external variables played a role in this discernibility: 1) Scan Direction, 2) patient position, and 3) machine settings. The scan direction in which the CT scan progresses can influence the image's appearance, especially when considering tissue boundaries and interfaces. Moreover, even minor deviations in patient positioning can lead to variations in image quality. This becomes particularly pertinent when considering orthopedic cases where precise positioning can influence the visualization of fractures or deformities. Furthermore, diverse CT machine settings, encompassing parameters like tube current, rotation speed, and scan duration, can significantly affect the resultant image. These variations might be introducing nuances in the image that our deep learning models haven't yet learned to reproduce accurately.

Lastly, our technique, utilizing cGANs for enhancing CT image resolution, holds significant potential for broader application in various medical fields. Firstly, many local clinics rely on older CT machines, which often produce lower-quality images. Retaking CT scans to improve image quality poses an additional risk due to increased radiation exposure, making our cGAN-based enhancement protocol a safer alternative. Secondly, the development of patient-specific implants demands high-resolution CT images to ensure precision and customization. Our technique can upgrade lower-quality scans to the required level of detail, thereby supporting more accurate implant fabrication. Lastly, the emergence of C-arm CT imaging in surgical settings is a promising advancement, but the current image quality often falls short of clinical needs. Enhancing these images using our cGANs protocol could greatly improve their utility in surgical planning and execution, providing clearer, more detailed visual information. This adaptability of our cGAN-based enhancement method across different scenarios underlines its potential as a transformative tool in medical imaging, particularly in situations where image quality is paramount yet constrained by existing equipment limitations.

Our results, while promising, also point towards areas of potential improvement. Increasing the sample size will expose our model to a wider array of CT images with varied intricacies. This could refine its learning and allow it to better approximate true 1 mm slices. Additionally, iteratively refining our deep learning protocol by incorporating feedback from clinical evaluations, and perhaps integrating newer neural network architectures, can further bridge the resolution and quality gap.

There were some limitations. Firstly, the study utilized a small patient pool from a single-center, which might introduce bias and limit the generalizability of the results. A multi-center study with a larger sample size might provide more robust and generalized conclusions. Additionally, while conditional GANs have showcased promising results, there are inherent challenges associated with GANs, such as mode collapse and training instability. Third, it's essential to acknowledge that while the enhanced images were of higher resolution than the 3 mm slice, achieving the quality of true 1 mm slices remains a challenge. Factors like scan direction and diverse machine settings introduce variations that a deep learning model might struggle with. Addressing these challenges in future iterations will ensure consistent and reliable image enhancements. Lastly, we had the lack of direct comparison between virtually generated 3 mm images and real CT-acquired 3 mm images. While our primary focus was on the enhancement of CT image resolution using conditional GANs, future research could benefit from including such comparisons to further validate the accuracy and effectiveness of the virtual generation process. Despite these limitations, the promising results presented here hold significant implications for orthopedics and radiology. With enhanced image quality, surgeons can better understand the anatomy, leading to improved surgical outcomes. Furthermore, the proposed method can reduce the need for repeated scans, reducing radiation exposure and healthcare costs.

## Conclusions

In our study aimed at refining CT imaging resolution using conditional Generative Adversarial Networks (cGANs), we assessed the potential of using this advanced computational technique to enhance the resolution of CT scans while maintaining patient safety. Our quantitative findings indicated that our proposed method, which emphasizes learning the difference value between the 3 mm input and the 1 mm corresponding image, consistently yielded superior image resolution. Moreover, qualitative evaluations by orthopedic surgeons further confirmed that images produced by this method were closer in quality to 1 mm slice images, reinforcing its clinical applicability. Consequently, conditional GANs offer a promising pathway for achieving improved CT image resolution without compromising patient safety, addressing concerns over radiation exposure and re-examinations, and ultimately aiding in more precise diagnostics and therapeutic interventions in orthopedics.

## Author Contributions

**Conceptualization:** Hyojune Kim, Kyung-Nam Kim.

**Data curation:** Sang-Il Oh, Young Ho Shin.

**Investigation:** Hyojune Kim, Ji-Soo Keum.

**Methodology:** Hyojune Kim, Seung Min Ryu, Ji-Soo Keum.

**Supervision:** In-Ho Jeon, Kyoung Hwan Koh.

**Validation:** Kyoung Hwan Koh.

**Writing – original draft:** Hyojune Kim.

**Writing – review & editing:** Kyoung Hwan Koh.

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
