## [Decision Letter · Decision Letter 0]

10 Jun 2024

PONE-D-24-07461Clinical validation of enhanced CT imaging for distal radius fractures through deep learning based interpolation and thickness reductionPLOS ONE

Dear Dr. Koh,

Thank you for submitting your manuscript to PLOS ONE. After careful consideration, we feel that it has merit but does not fully meet PLOS ONE’s publication criteria as it currently stands. Therefore, we invite you to submit a revised version of the manuscript that addresses the points raised during the review process.

We look forward to receiving your revised manuscript.

Kind regards,

Ngie Min Ung

Academic Editor

PLOS ONE

Journal Requirements:

A clean copy of the edited manuscript (uploaded as the new *manuscript* file)”.

4. We note that Figure(s) 4 and 7 in your submission contain copyrighted images. All PLOS content is published under the Creative Commons Attribution License (CC BY 4.0), which means that the manuscript, images, and Supporting Information files will be freely available online, and any third party is permitted to access, download, copy, distribute, and use these materials in any way, even commercially, with proper attribution. For more information, see our copyright guidelines: http://journals.plos.org/plosone/s/licenses-and-copyright.

a. You may seek permission from the original copyright holder of Figure(s) 4 and 7 to publish the content specifically under the CC BY 4.0 license. 

Reviewers' comments:

Reviewer's Responses to Questions

**Comments to the Author**

1. Is the manuscript technically sound, and do the data support the conclusions?

Reviewer #1: Yes

Reviewer #2: Partly

2. Has the statistical analysis been performed appropriately and rigorously? 

Reviewer #1: Yes

Reviewer #2: Yes

3. Have the authors made all data underlying the findings in their manuscript fully available?

Reviewer #1: Yes

Reviewer #2: Yes

4. Is the manuscript presented in an intelligible fashion and written in standard English?

Reviewer #1: Yes

Reviewer #2: Yes

5. Review Comments to the Author

Reviewer #1: Tha manuscript focuses on new methodologies to present high resolution (z-axis) images from low resolution images using neural networks. The methods are sound and logical, presented in an intelligible fashion. It is a very interesting read indeed. However, there are minor issues that require some additional information that it is not possible for me to recommend for an acceptance, at least on its current state. Please consider the following comments:

Please use "xxx mm" not "xxxmm" as this is more grammatically acceptable in writing physical measurement units.

In the introduction, there was mention of risk regarding patient dose, but this is not discussed at all throughout the manuscript. It is obvious that the dose (at least the dose length product, DLP) of 3 mm images will be lower than 1 mm. So generating 1 mm image from 3 mm acquisition will be beneficial at this point that deserves to be mentioned/discussed to support the idea of this paper.

A first thought was that the 1 mm image is generated from 3 mm images through two-times exposure (1 mm and 3 mm) of the subject as pairs with the accuracy of generated 1 mm being checked against the real 1 mm along the process of evaluation, but the case in the manuscript seems to be the other way around. Is it possible to disclose the reason? I suspect this is ethically-motivated (?)

Since the 3 mm images are generated (pg 7, lines 122-124), is there any mechanism discussed to verify between generated 3 mm images and real (CT-acquired) 3 mm images?

Reviewer #2: The paper presents an evaluation on the potential of conditional GANs in achieving superior CT image resolution without compromising patient safety in orthopedic imaging. The paper is well-written (with few grammatical errors) and can be accepted with few revisions.

Here are the suggestions to be considered by the authors:

1. The title should be revised. The term 'conditional Generative Adversarial Networks (cGAN)' should be included in the title to clarify the advanced method being discussed.

2. Please revise the range '3 mm-1 mm', it should be written from small to large, 1 mm - 3 mm. Add space between the value and unit (eg. 1 mm). Please correct the range throughout the text.

3. In methodology, the metrics part - please explain in details the calculation of mean squared error (MSE), and structural

similarity index (SSIM). Any related equation?

4. In methodology, the qualitative evaluation part - two orthopedic surgeons were included, but no radiologist was included as observers, justify? The criteria for the evaluation should be discussed in details, any specific anatomical or bony structure that been observed for spatial resolution evaluation? Please explain in details the criteria for each grade 1 to 4.

4. Please recheck the arrangement of figures and figures legends. It seems not matching to each other. Figures in last page should be label or indicated with arrow(s) to show the differences between each images. If possible, the authors can provide the zoom version on the structures (that shows improvement).

5. In results, the figure (A-B) that shows the histogram of the evaluation of the clinicians, it is suggested to add *label or notes that described the grade 1 to 4 for better understanding. Justify the significance of the figure B, for the choice for higher resolution image (why need this graph?), because from the grading, it seems like acquired images by 1 mm thickness appear to be superior among all.

6. In conclusion, please revise the sentence 'consistently yielded superior image resolution compared to traditional approaches'. What does it means by 'traditional methods'?. From the grading, it seems like images acquisition by 1 mm thickness appear to be superior among all.

6. PLOS authors have the option to publish the peer review history of their article (what does this mean?). If published, this will include your full peer review and any attached files.

Reviewer #1: No

Reviewer #2: No

---

## [Author Response · Author response to Decision Letter 0]

26 Jun 2024

Response to Reviewers

Reviewer Comments:

Reviewer #1:

1. Please use "xxx mm" not "xxxmm" as this is more grammatically acceptable in writing physical measurement units.

Response: Thank you for highlighting these key concerns. I revised it all through manuscript. 

2. In the introduction, there was mention of risk regarding patient dose, but this is not discussed at all throughout the manuscript. It is obvious that the dose (at least the dose length product, DLP) of 3 mm images will be lower than 1 mm. So generating 1 mm image from 3 mm acquisition will be beneficial at this point that deserves to be mentioned/discussed to support the idea of this paper.

Response: Thank you for your comments. We mentioned that the need for high quality of images by 1 mm slice CT get higher. Therefore, additional CT examination could be required for the patients who had 3 mm CT. It could induce the radiation exposure risk. It was mentioned in discussion (Line 417-420). However, we did not measure directly the exposed radiation dose from 3 mm CT and 1 mm CT. Therefore, we did not discuss the radiation dose. 

3. A first thought was that the 1 mm image is generated from 3 mm images through two-times exposure (1 mm and 3 mm) of the subject as pairs with the accuracy of generated 1 mm being checked against the real 1 mm along the process of evaluation, but the case in the manuscript seems to be the other way around. Is it possible to disclose the reason? I suspect this is ethically-motivated (?)

Response: Thank you for your insightful comment. We understand the potential confusion regarding the generation of 1 mm images from 3 mm images. In our study, we used paired 3 mm-1 mm CT data from patients who had undergone a single CT examination at 1 mm slice thickness. The 3 mm images were virtually derived from these 1 mm scans. This approach was chosen to minimize patient exposure to radiation, adhering to ethical standards that prioritize patient safety by reducing the need for multiple exposures. We will clarify this in the manuscript to emphasize the ethical considerations and the method used to generate the 3 mm images from the initial 1 mm scans. This method ensures that patients are not subjected to unnecessary additional radiation, aligning with best practices in medical imaging.

Revisions to Manuscript:

Line 120-123: In this study, we utilized paired 3 mm-1 mm CT data from patients who underwent a single CT examination at 1 mm slice thickness. The 3 mm images were virtually generated from these 1 mm scans to minimize radiation exposure, adhering to ethical standards and prioritizing patient safety.

4. Since the 3 mm images are generated (pg 7, lines 122-124), is there any mechanism discussed to verify between generated 3 mm images and real (CT-acquired) 3 mm images?

Response: Thank you for pointing out this important aspect. In our study, the 3 mm images were virtually generated from the 1 mm CT images. We did not perform a direct comparison between these virtually generated 3 mm images and real (CT-acquired) 3 mm images. This approach was primarily focused on demonstrating the feasibility of using conditional GANs to enhance the resolution of CT images. We acknowledge that verifying the similarity between virtually generated 3 mm images and real 3 mm images could further validate our approach. We will discuss this limitation in our manuscript and propose it as a direction for future research.

Revisions to Manuscript:

Line 446-450: Lastly, we had the lack of direct comparison between virtually generated 3 mm images and real CT-acquired 3 mm images. While our primary focus was on the enhancement of CT image resolution using conditional GANs, future research could benefit from including such comparisons to further validate the accuracy and effectiveness of the virtual generation process. 

Reviewer #2: 

1. The title should be revised. The term 'conditional Generative Adversarial Networks (cGAN)' should be included in the title to clarify the advanced method being discussed.

Response: Thank you for comments. We revised it, as ‘Clinical validation of enhanced CT imaging for distal radius fractures through conditional Generative Adversarial Networks (cGAN)’. 

2. Please revise the range '3 mm-1 mm', it should be written from small to large, 1 mm - 3 mm. Add space between the value and unit (eg. 1 mm). Please correct the range throughout the text.

Response: We revised. 

3. In methodology, the metrics part - please explain in details the calculation of mean squared error (MSE), and structural similarity index (SSIM). Any related equation?

Response: Thank you for comments. The referred metrics (MSE, SSIM) were generally used for evaluating the image generating experiments. Therefore, we just used them by reference article, which was ‘Sun Y, Pan B, Li Q, Wang J, Wang X, Chen H, et al. Clinical ultra-high resolution CT scans enabled by using a generative adversarial network. Med Phys. 2023;50(6):3612-22.‘.

4. In methodology, the qualitative evaluation part - two orthopedic surgeons were included, but no radiologist was included as observers, justify? The criteria for the evaluation should be discussed in details, any specific anatomical or bony structure that been observed for spatial resolution evaluation? Please explain in details the criteria for each grade 1 to 4.

Response: Thank you for your valuable feedback. We appreciate the opportunity to clarify and improve our methodology section. We have revised the methodology section to include detailed criteria for the evaluation. The surgeons focused on specific anatomical and bony structures (carpal bones and distal radius, joint space), such as cortical bone margins, fracture lines, and trabecular bone patterns. 

Revisions to Manuscript: Line 274-285

The grading criteria have been explicitly defined as follows:

- Grade 1: Excellent resolution with clear visualization of fine anatomical details for carpal bone and distal radius, including sharp cortical bone margins, fracture lines and distinct trabecular patterns.

- Grade 2: Good resolution with minor blurring but still adequate for clinical assessment of anatomical details for carpal bone and distal radius, but not exact for the fracture lines or trabecular patterns.

- Grade 3: Fair resolution with noticeable blurring, affecting the visibility of finer details, such as carpal bone and fracture patterns, but still allowing for basic clinical evaluation.

- Grade 4: Poor resolution with significant blurring, making it difficult to distinguish anatomical structures and unsuitable for clinical use for evaluating the fractures or anatomical structures.

5. Please recheck the arrangement of figures and figures legends. It seems not matching to each other. Figures in last page should be label or indicated with arrow(s) to show the differences between each image. If possible, the authors can provide the zoom version on the structures (that shows improvement).

Response: Thank you for comments. We revised the figs 7 and 8. 

6. In results, the figure (A-B) that shows the histogram of the evaluation of the clinicians, it is suggested to add *label or notes that described the grade 1 to 4 for better understanding. Justify the significance of the figure B, for the choice for higher resolution image (why need this graph?), because from the grading, it seems like acquired images by 1 mm thickness appear to be superior among all.

Response: Thank you for comments. Even though our method could nearly enhance the 3 mm images into the quality, the extent of 1 mm original version, the grading showed clearly difference between 1 mm image and generating images from Method 1 and 2. And the grading between Method 1 and 2 was not significantly different. As we want to figure out the superiority between Method 1 and 2, we try to choose from two generated images, which was more similar to 1 mm image by clinicians. 

7. In conclusion, please revise the sentence 'consistently yielded superior image resolution compared to traditional approaches'. What does it means by 'traditional methods'?. From the grading, it seems like images acquisition by 1 mm thickness appear to be superior among all.

Response: Thank you for comments. We revised the sentence. ‘Our quantitative findings indicated that our proposed method, which emphasizes learning the difference value between the 3 mm input and the 1 mm corresponding image, consistently yielded superior image resolution.’

---

## [Decision Letter · Decision Letter 1]

23 Jul 2024

Clinical validation of enhanced CT imaging for distal radius fractures through conditional Generative Adversarial Networks (cGAN)

PONE-D-24-07461R1

Dear Dr.
Kyoung Hwan Koh,

We’re pleased to inform you that your manuscript has been judged scientifically suitable for publication and will be formally accepted for publication once it meets all outstanding technical requirements.

Kind regards,

Ngie Min Ung

Academic Editor

PLOS ONE

Additional Editor Comments (optional):

Reviewers' comments:

Reviewer's Responses to Questions

**Comments to the Author**

1. If the authors have adequately addressed your comments raised in a previous round of review and you feel that this manuscript is now acceptable for publication, you may indicate that here to bypass the “Comments to the Author” section, enter your conflict of interest statement in the “Confidential to Editor” section, and submit your "Accept" recommendation.

Reviewer #1: All comments have been addressed

Reviewer #2: All comments have been addressed

2. Is the manuscript technically sound, and do the data support the conclusions?

Reviewer #1: Yes

Reviewer #2: Yes

3. Has the statistical analysis been performed appropriately and rigorously? 

Reviewer #1: Yes

Reviewer #2: Yes

4. Have the authors made all data underlying the findings in their manuscript fully available?

Reviewer #1: Yes

Reviewer #2: Yes

5. Is the manuscript presented in an intelligible fashion and written in standard English?

Reviewer #1: Yes

Reviewer #2: Yes

6. Review Comments to the Author

Reviewer #1: The authors have addressed all the comments. It is now appropriate for me to recommend for an acceptance.

Reviewer #2: The authors have addressed all the comments and suggestions provided. The responses and the subsequent revisions have adequately addressed reviewers' concerns, and the manuscript is significantly improved.

I am pleased to recommend the acceptance of the revised manuscript for publication.

7. PLOS authors have the option to publish the peer review history of their article (what does this mean?). If published, this will include your full peer review and any attached files.

Reviewer #1: No

Reviewer #2: No

---

## [Editor Report · Acceptance letter]

8 Aug 2024

PONE-D-24-07461R1 

PLOS ONE

Dear Dr. Koh, 

I'm pleased to inform you that your manuscript has been deemed suitable for publication in PLOS ONE. Congratulations! Your manuscript is now being handed over to our production team.

Kind regards, 

on behalf of

Dr. Ngie Min Ung 

Academic Editor

PLOS ONE